# Health Literacy and Medication Adherence Among Low-Income Older Adults in the Inner Center of Portugal: A Questionnaire-Based Study

**DOI:** 10.3390/healthcare13202560

**Published:** 2025-10-11

**Authors:** Carla Perpétuo, Ana I. Plácido, Alexandra Monteiro, Ramona Mateos-Campos, Maria Teresa Herdeiro, Fátima Roque

**Affiliations:** 1Local Health Unit of Guarda (ULSG), 6301-857 Guarda, Portugal; carladicarol@ipg.pt; 2BRIDGES—Biotechnology Research, Innovation and Design for Health Products, Polytechnic University of Guarda, 6300-559 Guarda, Portugal; 3Faculty of Pharmacy of the University of Salamanca, Campus Miguel de Unamuno, Calle Lic. Méndez Nieto, s/n, 37007 Salamanca, Spain; 4Area of Preventive Medicine and Public Health, Department of Biomedical and Diagnostic Sciences, University of Salamanca, 37008 Salamanca, Spain; 5Department of Medical Sciences, Institut e of Biomedicine (iBiMED-UA), University of Aveiro, 3810-193 Aveiro, Portugal

**Keywords:** health literacy, medication adherence, older adults, socioeconomic factors

## Abstract

**Introduction:** Health literacy (HL) is a key determinant of health outcomes, particularly among vulnerable populations such as low-income older adults. Limited HL is associated with poor understanding of medication instructions and lower medication adherence, which can compromise therapeutic success. This study aims to assess the levels of HL and medication adherence among low-income older adults and to analyse the relationship between HL, medication adherence, and other determinants. **Methods**: A cross-sectional study was conducted among low-income older adults in Portugal’s Beira and Serra Estrela Region. HL was assessed using the European Health Literacy Survey Questionnaire (HLS-EU-PT), and medication adherence was measured with the Adherence to Treatment Measure (MAT) scale. Descriptive statistics, Spearman correlations, ordinal logistic regression, and linear regression were used to analyse associations between HL, adherence, and sociodemographic and health factors. **Results:** Of 196 participants, most had problematic or inadequate HL (87.8%). Medication adherence was relatively high (mean = 5.37, SD = 0.54). HL was positively associated with medication adherence (R = 0.260, *p* < 0.001), education (R = 0.277, *p* < 0.001), and ability to pay expenses (R = 0.235, *p* = 0.002) and negatively with age (R = −0.179, *p* = 0.019), poor health status (R = −0.237, *p* = 0.002), and difficulty affording medication (R = −0.389, *p* < 0.001). Completion of the third cycle of primary education predicted higher HL (OR = 1.939, 95% CI: 0.088–3.790, *p* = 0.040); the ability to pay expenses predicted better adherence (B = 0.101, 95% CI: 0.014–0.187, *p* = 0.022). **Conclusions**: Low HL remains a significant barrier among low-income older adults in Portugal, despite generally high medication adherence. Education and financial stability are key determinants to HL and adherence. Interventions should integrate HL promotion with efforts to reduce socioeconomic barriers and support medication management in the vulnerable populations.

## 1. Introduction

Health literacy (HL) is essential for ensuring positive health outcomes, particularly in vulnerable populations such as older adults with low-income backgrounds [1,2,3]. HL is defined as the extent to which an individual obtains, processes, and understands basic health information and services related to healthcare. HL is also recognized as a fundamental person-level determinant of medication adherence [4,5]. Limited HL has been related to a poor understanding of medication instructions, limited engagement in healthcare services, and lower medication adherence [6].

A global perspective on HL reveals persistent and significant disparities across world regions, impacting healthcare utilization, health outcomes, and well-being. Recent statistics from 2024 show that globally, nearly half of adults demonstrate inadequate HL, with high rates of limited HL in both developed and developing nations [7]. Within the European Union, population-based surveys parallel global trends, reporting significant proportions of adults with problematic or inadequate HL and highlighting at-risk groups such as older adults and those with lower socioeconomic status [1,8].

The European Health Literacy Survey Questionnaire (HLS-EU-Q) was designed to look at a more comprehensive, multidimensional form of HL across several countries [1,8]. The findings from this European survey reported that nearly half of adults had inadequate or problematic HL that contributed to inappropriate health behaviors, poorer health outcomes, and increased healthcare costs [1,8]. In 2023, there was an overall estimate of 24.1% of the Portuguese population who were 65 or older [9]. Older adults have several chronic conditions that require complex medication regimens, face financial burdens, and have limited access to healthcare services (especially when alone and/or in rural areas) [10]. The findings of the translation and validation of HLS-EU-Q to the Portuguese population stated that approximately 61% of the population had problematic or inadequate HL, which is a demonstrated need for improvement of HL and reflects the critical need for public health intervention strategies [2]. Recent research has repeatedly reaffirmed these concerns: research conducted in a surgical oncology unit found that patients had problematic HL levels [11]; another study conducted in a heart failure unit was consistent with this finding, where two-thirds of patients had low HL, especially those with lower income, and poor social support or greater geographical distance from healthcare services [12]. These findings highlight significant gaps in HL between different populations and settings in Portugal, pointing to the need for targeted interventions to tackle disparities [13].

The Health Literacy Population Survey 2019–2021 showed that 45.3% of the Portuguese population has problematic or inadequate HL, with the most vulnerable being older adults and low-income individuals [14], and confirmed that there are significant disparities in HL, especially for older adults, those with lower levels of education, and those with lower income [13]. These findings highlight the need to address the social determinants of health and to promote strategies to mitigate inequities in access to health information and services.

Medication adherence is crucial to achieving therapeutic success [15]. Research indicates that individuals with low HL are likely to misinterpret medication instructions, miss medication schedules, or discontinue treatments [16,17]. These issues are pronounced among older adults with multiple medications, cognitive impairments, and social vulnerabilities [18].

Recent research shows a significant association between HL and medication adherence in older adults; however, most of the literature in this area has focused on general populations of older adults or groups with specific chronic diseases and not on low-income older adults [19,20,21], leaving a gap to fill in the literature in regards to these populations and assessing this relationship. Furthermore, there is limited exploration of socioeconomic and contextual factors affecting these vulnerable groups and a scarcity of longitudinal and intervention studies that could inform tailored, culturally sensitive strategies to enhance both HL and medication adherence. Therefore, conducting research with the low-income older adult population is relevant and necessary to develop and study group-specific interventions for low-income older adults.

This study aims to determine the level of HL and medication adherence among low-income older adults, analyse the relationship between HL and medication adherence, and identify the main sociodemographic, health, and economic factors associated with these outcomes.

## 2. Materials and Methods

### 2.1. Study Design

This is a cross-sectional study aimed at assessing the level of HL and medication adherence among low-income older adults using validated versions of the European Health Literacy Survey Questionnaire (HLS-EU-PT) [2] and the Adherence to Treatment Measure (MAT) [22] scale.

### 2.2. Ethics Considerations

The study received ethics approval from the Ethics Committee of the Polytechnic Institute of Guarda (Nº. 2/2023 of 24 April 2023). All participants provided written informed consent before enrolment, and data were pseudo-anonymized to ensure confidentiality, following the Helsinki Declaration and the General Data Protection Regulation (GDPR).

### 2.3. Participants

The participants were low-income older adults who lived in the Intermunicipal Community of the Beira and Serra Estrela Region (CIMRBSE), a Nomenclature of Units for Territorial Statistics—NUTS III (classification used by Eurostat to organize and compare regional statistics across Europe) [23]. These low-income older adults were recruited by their municipalities of residence, which introduced the study objectives, asked if they wished to participate, and provided their contact information to the research team after signing an informed consent form.

Inclusion criteria were age 65 years or older, low income, ability to provide informed consent to participate in the study, and ability to complete the questionnaires independently. Exclusion criteria include cognitive impairment, institutionalization, or dependency.

THE CIMRBSE is a predominantly rural and aging region, with a low population density, a high proportion of older adults, and a declining population trend. It includes 15 municipalities, but only 11 decided to participate in the study [24]. The target population consisted of 55,358 older adults residing in the 11 municipalities of CIMRBSE. According to the most recent data, 19.9% of older adults in the Centro region are at risk of poverty after social transfers, which corresponds to an estimated 11015 low-income older adults in the study area.

The criteria used to define “low income” in this study are based on the eligibility for the Solidarity Supplement for the Elderly (CSI) and related social support thresholds in Portugal, which consider factors such as annual and monthly income limits relative to the Social Support Index (IAS), residency status, and pension or disability benefits, as detailed in Perpétuo et al. (2025) and aligned with national social security regulations [25].

### 2.4. Data Collection

After obtaining informed consent from the participants, the research team provided a schedule for them to visit the parish councils, where the researchers administered the questionnaires in person. Before completing the questionnaire, participants were thoroughly briefed on the study objectives and given the informed consent form. All data were collected in a session lasting approximately 30 to 45 min, between February and November 2024.

The questionnaire includes 4 sections: Section A (sociodemographic characterization—age, gender, marital status, education, income), Section B (health status and behaviors—self-perceived health, chronic conditions, lifestyle habits), Section C (HL assessment using the HLS-EU-PT questionnaire), and Section D (medication adherence evaluation using the MAT scale).

### 2.5. Questionnaires

Two validated self-administered questionnaires were used to collect data: the HLS-EU-PT, Portuguese version, and the MAT scale.

#### 2.5.1. HLS-EU-PT Questionnaire

To assess the level of HL among participants, we applied the HLS-EU-PT questionnaire (2). This instrument was translated, culturally adapted, and validated for the Portuguese population by Pedro et al. (2016) [2] and has demonstrated strong psychometric properties (Cronbach’s alpha > 0.90 across subscales). The HLS-EU-PT consists of 47 items and evaluates HL across three domains: healthcare, disease prevention, and health promotion, and four cognitive processes: accessing, understanding, appraising, and applying health information.

Each item was scored using a 4-point Likert scale (from “very easy” to “very difficult”), and total scores were transformed into a metric scale ranging from 0 to 50. Based on cut-off points established by the original European Health Literacy Survey Consortium, individuals are categorized into four HL levels: inadequate (≤25), problematic (26–33), sufficient (34–42), and excellent (43–50) (1).

#### 2.5.2. MAT Scale

To assess medication adherence, we applied the MAT scale [22], translated and psychometrically validated to ensure cultural and linguistic appropriateness for the Portuguese population. It is a self-report instrument composed of seven items evaluating the frequency of non-adherent behaviours. Participants responded using a 6-point Likert scale ranging from 1 (“Always”) to 6 (“Never”). Higher scores reflect higher levels of treatment adherence. The total adherence score was calculated by averaging the scores across all items.

### 2.6. Data Analysis

Descriptive statistics, including frequencies, percentages, means, medians, and standard deviations, were generated using IBM SPSS v29.0.The “Don’t know/don’t answer” was considered a missed value.

In the HLS-EU-PT, questionnaires with more than 20% missing responses (fewer than 38 items answered) were excluded from the analysis, as recommended in the original HLS-EU methodology (1), and missing values were treated as such and were ignored in the analysis. The mean of the valid items per older adult was calculated and then converted to a scale from 0 to 50.

Regarding the MAT scale [22] the adherence level was obtained by summing up the values from the seven questions and dividing the sum by the number of questions. Cronbach’s alpha was calculated to measure the internal consistency of the applied scales.

Non-parametric tests were used because the variables of this study did not follow a normal distribution (*p* < 0.001). The Spearman Correlation value was obtained for quantifying associations between variables, continuous or ordinal. Ordinal logistic regression was used to predict HL level because it is a categorical, ordinal, and non-continuous variable, and linear regression was used to predict adherence levels, a continuous variable.

## 3. Results

### 3.1. Sample Characterization

A total of 196 individuals aged 65 years and older participated in the study, with a median age of 71 years, ranging from 65 to 95 years (Table 1). The majority were female (66.3%) and were 65 to 74 years old (69.4%). Most participants were married (57.1%) and lived with others (67.3%).

Health data showed that 91.8% did not smoke, and 69.4% did not consume alcohol. A majority (62.2%) reported practicing physical activity. Educational levels were generally low, with 73.5% having completed only up to the second cycle of primary education. Most were retired (79.6%).

Regarding the income, 69.9% found it difficult or very difficult to cover all expenses by the end of the month, and 53% reported difficulty affording medications. The majority (75.5%) had a monthly income below EUR 800.

Nearly 96% of older adults reported having a chronic disease or disability, and 70.4% stated that health problems limited their daily activities. Public healthcare was the main system used (94.9%), although 47.4% reported difficulty in accessing their assigned doctor. In the previous year, 66.3% used emergency services, 92.9% had at least one primary care consultation, and 30.6% were hospitalized (Table 2).

### 3.2. Health Literacy

Out of the 196 questionnaires administered, 172 (87.8%) were valid for HL analysis according to HLS-EU-PT completeness criteria. The mean general HL score was 28.07 (SD = 5.87), with domain-specific scores averaging 27.11 (SD = 7.00) for healthcare, 29.11 (SD = 6.90) for disease prevention, and 28.01 (SD = 6.33) for health promotion.

The General Health Literacy scale and the domains of Healthcare and Disease Prevention show good to excellent reliability (Cronbach’s alpha > 0.7), and the Health Promotion domain has low reliability (Cronbach’s alpha = 0.520) (Table 3).

Figure 1 shows the distribution of participants across the different health literacy levels. Across all domains, most older adults fall into the problematic category, especially in General Health Literacy level.

### 3.3. Medication Adherence

Among the 196, only 193 older adults completed the MAT scale, which demonstrates a Cronbach’s alpha = 0.689. This suggests that the items are reasonably correlated. The overall mean adherence score was 5.37 (SD = 0.54), with scores ranging from 3.43 to 6.00 and a median of 5.43, indicating a generally high level of adherence to prescribed medication regimens (Table 4).

Analysis of responses to the seven items of the MAT scale (Table 5) revealed that the most frequently reported non-adherence behaviours were: forgetting to take medications (sometimes: 33.2%, never: 37.2%), being careless about medication timing (sometimes: 32.1%, never: 44.9%), and interrupting therapy due to running out of medicines (sometimes: 21.9%, never: 58.7%).

Intentional non-adherence behaviours, such as stopping medication after feeling better or worse, or self-adjusting doses, were less frequent. Therefore, 17.3% of participants have stopped their medication because they felt better, 18.4% have taken extra doses when feeling worse, and 21.9% have discontinued their medication for reasons unrelated to medical appointments.

### 3.4. Health Literacy and Medication Non-Adherence Related Factors

HL level is positively associated with medication adherence, educational level, and the ability to pay all monthly expenses, while it is negatively associated with age, perceived health status, hospitalization, and difficulty affording medication.

Medication adherence is positively associated with educational level and negatively associated with perceived health status and difficulty affording medication. These correlations are statistically significant and suggest that socioeconomic and health-related factors influence HL and medication adherence. Improving education and financial stability may contribute to better HL and treatment adherence (Table 6).

In the ordinal logistic regression analysis, only 3rd cycle of primary education showed a significant association with higher levels of HL (B = 1.393; *p* = 0.040). None of the other variables analysed showed a significant association with the outcome. In the linear regression analysis, the “ability to pay expenses” variable is associated with an average increase of 0.101 in the medication adherence score (Table 7).

## 4. Discussion

This study examined HL and medication adherence of low-income older adults in Portugal, as well as their association and influencing factors. The results demonstrated a low mean general health literacy score, with the majority of participants classified as having problematic or inadequate HL, which may affect their ability to manage health and navigate healthcare systems [1]. The proportion of participants classified as having sufficient or excellent HL is low, suggesting that the majority of older adults have difficulty understanding, processing, or using health information more effectively. The HLS-EU-PT tool allows a multidimensional assessment of perceived HL and the classification of individuals into different levels of need for intervention to target strategies to improve health outcomes, namely medication adherence.

These findings are consistent with previous Portuguese and European studies, which report that over 60% of older adults and those with lower socioeconomic status have limited HL [2,8] and reinforce the persistent disparities observed in the Portuguese context, where older adults and those with lower income and lower education levels are disproportionately affected by limited HL [14].

The General Health Literacy scale and its Healthcare and Disease Prevention domains demonstrated good to excellent internal consistency, supporting their reliability. However, the Health Promotion domain had low internal consistency, which could suggest that the items in this domain may not be consistently measuring the same construct in the population studied. This could be due to participants misunderstanding some items, lower educational level, or maybe some of the questions were not culturally appropriate for this specific context. Therefore, results related to this domain should be interpreted with caution. For future studies, they may benefit from a qualitative review of the items, statistical reporting related to inter-item correlation, as well as modifications or removal that proved to be problematic. Further, it may be beneficial to add new questions that can relate better to the sociocultural context of low-income older adults [1,2,8,26].

Although very low HL levels, the sample had a high mean medication adherence score, with all older adults classified as adherent based on the MAT scale. It must also be noted that the answers were on self-perception, and the questions may often not be in line with reality. Research highlights that self-reported medicine adherence consistently overestimates true adherence due to social desirability bias, particularly among patients with lower education levels or anxiety symptoms [27]. Overall, the results suggest that although most participants remained largely adherent, occasional lapses were present due to forgetting to take a dose, taking it at the wrong time, or running out of medication.

The MAT scale’s internal consistency was moderate, which is acceptable for exploratory research. However, unintentional non-adherence, such as forgetting doses, carelessness with timing, and running out of medication, was common, while intentional non-adherence was less frequent. This finding is in line with international evidence identifying that difficulties in understanding medication instructions and, in some cases, managing complex management regimens, affect older adults’ perceptions of adherence, leading mainly to unintentional lapses [28,29,30].

Socioeconomic factors were also prevalent within the sample. Most participants reported a moderate level of difficulty covering their monthly expenses, and more than half had difficulty affording medication. Financial barriers compound HL and adherence issues due to limited funds, reducing access to health information, healthcare services, and essential medications. This is consistent with previous research that identifies financial strain as a very strong predictor of both low HL and poor medication adherence in older adults [31].

Similarly, this finding was aligned, reflecting that low HL is predominantly linked to challenges deciphering medication instructions and managing complicated schedules to follow complex treatments, contributing to unintentional lapses [30,32]. The predominance of unintentional non-adherence suggests that interventions aimed at simplifying medication regimens, improving patient education, and enhancing support systems for medication management will have more of an effect.

This study found that HL is positively correlated with medication adherence, indicating that individuals with better HL tend to adhere more closely to prescribed treatments. The relationship between HL and medication adherence is well documented in the literature. Zhang et al. (2013) [5], Miller (2016) [30] and Geboers et al. (2015) [32] demonstrate that patients with higher HL better understand medical instructions, recognize the importance of adherence, and can more effectively manage their medication, leading to better clinical outcomes. In Portugal, Pedro et al. (2016) [2] and Costa et al. (2023) [14] confirm that HL is a crucial determinant of adherence, especially among older adults. HL is a cross-cutting factor because it influences both the understanding of prescriptions and the ability to overcome financial and motivational barriers. Interventions to improve HL, such as those described by Wali et al. (2016) [31] and Schönfeld et al. (2020) [33], have shown a positive impact on adherence, particularly when tailored to the needs of vulnerable groups.

The educational level shows a strong positive correlation with HL, showing that education plays a big part in promoting appropriate health behaviors. Education is one of the most consistent predictors of HL, as noted by Pedro et al. (2016) [2] and Costa et al. (2023) [14]. Individuals with more years of schooling have a better understanding of health information and navigate the health system more easily.

Being able to pay all expenses at the end of the month is positively correlated with HL, while the capacity to afford medication had a negative correlation, suggesting that income can affect HL. Arriaga et al. (2022) [13] and Costa et al. (2023) [14] document that income can limit access to important information and health services, which can lead to lower HL.

Age is negatively correlated with HL, which suggests that HL declines with age. Studies such as Lima et al. (2024) [3] show that older adults often struggle with low HL due to declining cognitive skills, less access to digital information, and lower average schooling levels. Perceived health status and hospitalization are also negatively correlated with HL: people who perceive their health to be poorer or have been hospitalized tend to have less literacy, perhaps because they face more barriers to understanding and managing complex health information, as discussed by Ferreira et al. (2024) [11].

Across adherence, the positive relationship between educational level and medication adherence observed in this study is well-supported by literature that shows that more years of schooling generally lead to a better comprehension of medical recommendations, an improved understanding of prescriptions or doses, and ultimately greater autonomy over one’s health and well-being. Miller’s (2016) [30] meta-analysis reinforced that education, often correlated with HL, was one of the primary predictors of adherence, particularly in chronic disease contexts. In Portugal, Gomes et al. (2019) [15] also found that older adults with lower education levels had more difficulties in daily medication management. On the other hand, the negative relationship obtained between perceived health status and adherence indicates that individuals who perceive their health as poorer often take their medications less regularly. The literature suggests that declining health status may relate to increased therapeutic complexity, polypharmacy, and a higher number of symptoms, leading to decreased adherence. Additionally, a negative self-image might involve feelings of hopelessness or demotivation, resulting in lower commitment to treatment [33,34].

Besides health issues and therapeutic complexity, difficulty in accessing medicines, demonstrated by the negative association with adherence, is widely established as an important barrier, particularly for older and low-income populations. Research shows that lower income compromised the continuity of treatment, resulting in interruptions, dose rationing, or the cessation of treatment. Geboers et al. (2015) [32] highlight that, besides HL, income is a major reason for not following treatments.

The finding that only completion of the third cycle of primary education significantly predicts higher health literacy (HL) highlights a critical educational threshold within the older adult population in Portugal. This milestone appears to delineate a substantive divide in HL capacity, reflecting the wide variation in basic education levels typical among older adults in rural and low-income areas [25]. Individuals who do not reach this level are at greater risk of low HL, which can adversely affect their ability to manage health and adhere to medication regimens [2,14]. Also, the ability to pay expenses at the end of the month was the only significant predictor of better medication adherence, underscoring the importance of having a higher income, supporting the well-established link between education and health outcomes [34].

Given the high prevalence of chronic disease and functional limitations among the study population, targeted interventions to improve HL are clearly needed. Enhancing HL can not only boost medication adherence in low-income older adults but also reduce healthcare costs and improve quality of life. These interventions should adopt a multi-faceted approach, including education programs tailored to older adults, simplified communication by healthcare professionals, and community services to mitigate the impact of low HL on medication adherence [33]. Addressing financial barriers through policy and social support is also crucial. Digital health solutions, such as targeted e-learning programs, can enhance older adults’ self-efficacy and support medication management and communication with healthcare professionals [35].

This study’s findings highlight important clinical implications: the strong positive association between HL and medication adherence underscores the need for routine HL assessment, particularly among low-income older adults. Community pharmacists in Portugal play a vital role by offering medication management, personalized counseling, and HL support, which can significantly improve adherence and health outcomes [36,37,38]. Tailored HL interventions addressing the unique challenges faced by rural older adults are essential to overcome access barriers and health disparities. Furthermore, education level and financial capacity emerged as key predictors, emphasizing that social determinants must be integrated into clinical risk assessments and care plans. A multidisciplinary approach, including patient education, financial counseling, and pharmacist-led medication management, is recommended to optimize medication adherence and overall health outcomes in vulnerable elderly populations.

The results of this research demonstrate correlations at a single moment of study and do not allow us to understand the directionality or causation form of the relationship studied. Longitudinal and interventional research is required to identify the causal pathways and changes over time. The reliance on self-reported data introduces risks of recall bias relevant for measuring HL and medication adherence, and the HL questionnaire used in this study was reported to be challenging for participants to interpret and respond to accurately, which may have influenced the results. The sample is exclusively of low-income older adults living in a rural area of Portugal, the CIMRBSE. The sociodemographic and healthcare characteristics of rural populations are different than those in urban or more affluent areas. Therefore, these results may not reflect the experience of most Portuguese older adults, particularly those living in urban settings or with higher socioeconomic status. Future research could further enhance the external validity of the results through recruiting more diverse and larger sample sizes that reflect different geographic and socioeconomic situations in Portugal.

## 5. Conclusions

Low HL remains a significant barrier for low-income older adults in Portugal, although general medication adherence rates seem high. Education and financial stability are significant determinants of HL and adherence. The continuing non-adherence behaviors, whether unintentional, indicate a need to consider both cognitive determinants of adherence, as well as economic accessibility. Future interventions should adopt a holistic approach, integrating HL promotion with efforts to reduce socioeconomic barriers and support effective medication management in this vulnerable population. For example, tailored educational programs using simple language and culturally appropriate materials can enhance understanding and empower older adults. Social support initiatives, such as financial assistance for medication costs and improved access to healthcare services, especially for those in rural areas, are essential. Additionally, multidisciplinary collaborations involving healthcare professionals, social workers, and caregivers can ensure comprehensive support, while technology-based tools like medication reminders designed for older adults can improve adherence. Continuous evaluation and adaptation of these interventions, informed by patient feedback and outcome monitoring, will be crucial to address the complex and diverse needs of this population effectively.

## Figures and Tables

**Figure 1 healthcare-13-02560-f001:**
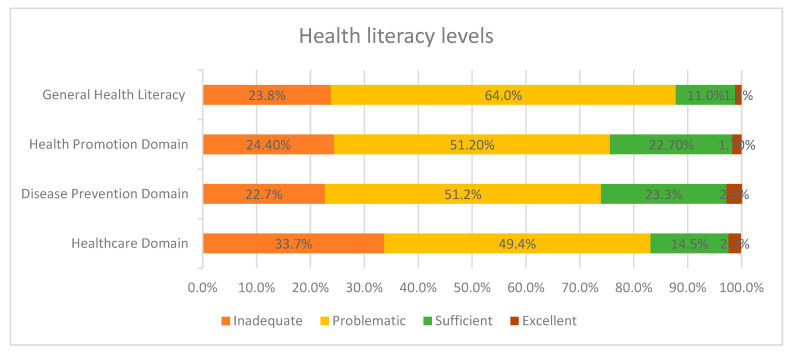
Health Literacy Levels.

**Table 1 healthcare-13-02560-t001:** Sociodemographic characterization.

	N (196)	%
Sociodemographic Characterization		
Age median (Q1–Q3)	71	(67–76)
65–74	136	69.4%
75–84	49	25%
≥85	11	5.6%
Gender		
Male	66	33.7%
Female	130	66.3%
Marital Situation		
Single	14	7.1%
Married	112	57.1%
Separated/divorced	27	13.8%
Widower	43	21.9%
Household living arrangement		
Living alone	64	32.7%
Living together	132	67.3%
Psychometric values	mean ± SD	min. and max
Height	159.9 ± 7.46	140–185
Weight	69.5 ± 13.1	35–130
Smoking		
Yes	16	8.2%
No	180	91.8%
Use of alcohol		
Yes	60	30.6%
No	136	69.4%
Practice of physical activity		
Yes	122	62.2%
No	74	37.8%
Educational level		
Level 0 (preschool)	17	8.7%
Level 1 (up to 2nd cycle of primary education)	144	73.5%
Level 2 (3rd cycle of primary education)	24	12.2%
Level 3 (high school)	11	5.6%
Employments status		
Working professionally	2	1.0%
Unemployed	6	3.1%
Retired	156	79.6%
Inability	15	7.7%
Housewife	8	4.1%
Inactive	6	3.1%
Other	3	1.5%
Pay all the expenses at the end of the month		
Easy or very easy	59	30.1%
Difficult or very difficult	137	69.9%
Afford medication		
Easy or very easy	92	47%
Difficult or very difficult	104	53%
Income		
Above EUR 500	62	31.6%
Between EUR 500 and EUR 800	86	43.9%
Between EUR 800 and EUR 1350	41	20.9%
Between EUR 1350 and EUR 1850	7	3.6%

**Table 2 healthcare-13-02560-t002:** Perceived health status and healthcare-related variables.

	N (196)	%
Health Status and Healthcare-Related Variables		
Self-perceived health status		
Good or very good	27	12.7%
Fair	124	63.3%
Bad or very bad	45	23%
Self-reported chronic disease or disability		
Yes	188	95.9%
No	8	4.1%
Health problems limit usual activities		
Yes	138	70.4%
No	58	29.6%
Type of health system		
Public	186	94.9%
Public and private	10	5.1%
Access to primary healthcare doctor		
Easy or very easy	103	52.6%
Difficult or very difficult	93	47.4%
Emergency service in the last year		
Yes	130	66.3%
No	66	33.7%
Primary healthcare medical consultation in the previous year, at least once		
Yes	182	92.9%
No	14	7.1%
Hospitalization in the last year		
Yes	60	30.6%
No	136	69.4%

**Table 3 healthcare-13-02560-t003:** Health Literacy Scores.

		Score (Mean ± SD)	Min.	Max.	Cronbach’s Alfa
General Health Literacy		28.07 ± 5.87	2.13	44.31	0.747
Health Literacy Domains	Healthcare	27.11 ± 7.00	2.08	47.92	0.886
	Disease prevention	29.11 ± 6.90	2.22	48.89	0.871
	Health promotion	28.01 ± 6.33	0	45.56	0.520

**Table 4 healthcare-13-02560-t004:** Low-income Older Adults’ Adherence Levels.

	Minimum	Maximum	Mean ± SD	Median
Adherence level	3.43	6	5.37 ± 0.54	5.43

**Table 5 healthcare-13-02560-t005:** Results from the application of the Adherence Treatment Measure (MAT) scale.

N (193)	Always % (N)	Almost Always % (N)	Often % (N)	Sometimes % (N)	Seldom % (N)	Never % (N)	Mean ± SD	Median
1. Have you ever forgotten to take the medicines for your illness?	-	1% (2)	7.7% (15)	33.2% (65)	19.4% (38)	37.2% (73)	4.85 ± 1.051	5
2. Have you ever been careless about the time you take your medicines?	-	3.1% (6)	6.1% (12)	32.1% (63)	12.2% (24)	44.9 (88)	4.91 ± 1.145	5
3. Have you ever stopped taking medicines for your illness because you felt better?	-	-	2% (4)	9.7% (19)	4.1% (8)	82.7% (162)	5.7 ± 0.731	6
4. Have you ever stopped taking the medicines for your illness on your own after feeling worse?	0.5% (1)	0.5% (1)	3.6% (7)	13.3% (26)	5.6% (11)	75% (147)	5.52 ± 0.952	6
5. Have you ever taken one or more pills for your illness on your own after feeling worse?	-	-	1.5% (3)	10.7% (21)	4.6% (9)	81.6% (160)	5.69 ± 0.727	6
6. Have you ever interrupted therapy for your illness because you have run out of medicines?	-	1% (2)	3.6% (7)	21.9% (43)	13.3% (26)	58.7% (115)	5.27 ± 0.995	6
7. Have you ever stopped taking your medicines for some reason other than a doctor′s appointment?	-	-	1% (2)	8.7% (17)	10.7% (21)	78.1% (153)	5.68 ± 0.676	6

**Table 6 healthcare-13-02560-t006:** Statistically significant Spearman correlation between health literacy, adherence level, sociodemographic characterization, health status, and income data.

Variables		Coefficient Value (R)	Significance (*p*-Value)
Health Literacy Level	Adherence level	0.260	<0.001
	Age	−0.179	0.019
	Educational level	0.277	<0.001
	Health perceived status	−0.237	0.002
	Hospitalization	−0.159	0.038
	Pay all the expenses at the end of the month	0.235	0.002
	Afford medication	−0.389	<0.001
Adherence level	Health perceived status	−0.169	0.028
	Educational level	0.323	<0.001
	Afford medication	−0.229	0.003

**Table 7 healthcare-13-02560-t007:** Predicted association between HL, medication adherence, and other variables.

Variables		OR (CI 95%)	*p*-Value
Health Literacy Level	3rd cycle of primary education	1.939 (0.088–3.790)	0.040
Adherence level	Pay all the expenses at the end of the month	0.101 (0.014–0.187)	0.022

OR—Odds Ratio; CI—Confidence Interval.

## Data Availability

In accordance with Ethics Committee approval and participants’ informed consent, data cannot be shared at the individual level. Only aggregated results, as reported in the manuscript, are available.

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
