# Peer review of "Health Literacy and Medication Adherence Among Low-Income Older Adults in the Inner Center of Portugal: A Questionnaire-Based Study"

_healthcare, 2025, doi:10.3390/healthcare13202560_

Round 1

Reviewer 1 Report

Comments and Suggestions for Authors

Health Literacy and Medication Adherence among Low-Income Older Adults in the Inner Centre of Portugal: A Questionnaire- Based Study

Comments

Abstract

Results: need to add the adequate statistical values to support the conclusion.

Methods

Ethics details is missing. Provide ethical approval number and where obtained ( Name of the institution)

For Questionnaire Please provide the scorings in a separate sub heading.

2.3.1. HLS-EU-PT Questionnaire

For this questionnaire have you done the reliability analysis I mean cornbach alpha value??

2.3.3. MAT Scale

For this questionnaire have you done the reliability analysis I mean cornbach alpha value??

 Missing??

Provide the details

Total sample included in the study need to be included in the methodology better to provide in flow chart

Clarify how did you obtain the iformed consent???

Data collection- Questionnaire administered online or how?? need to incorporate??

Under data collection what are the data collected I mean provide description of the data collection form example Section A demogrprahic details Age, gender etc..section B… Section c

Table 6,7,8---correct the font size inside the table explain what is OR, CI below the table for all the table accordingly

Discussion

Incorporate the CLinical implications

Add the limitations of the study under the discussion section

Conclusion

The following interventions written in brief. Try to elaboarte with examples  and how can we improve..

Future interventions should adopt a holistic

approach, integrating HL promotion with efforts to reduce socioeconomic barriers and

support effective medication management in this vulnerable population.

Best wishes

Author Response

Health Literacy and Medication Adherence among Low-Income Older Adults in the Inner Centre of Portugal: A Questionnaire- Based Study

Comment 1: Abstract

Results: need to add the adequate statistical values to support the conclusion.

Response 1: Thank you for pointing this out. We improved the result section in the abstract to add adequate statistical values to support the conclusion.

Line 30-36

“Of 196 participants, most had problematic or inadequate HL (87.8%). Medication adherence was relatively high (mean = 5.37, SD = 0.54). HL was positively associated with medication adherence (R = 0.260, p < 0.001), education (R = 0.277, p <0.001), and ability to pay expenses (R = 0.235, p = 0.002); and negatively with age (R = -0.179, p = 0.019), poor health status (R = -0.237, p = 0.002), and difficulty affording medication (R = -0.389, p < 0.001). Completion of the third cycle of primary education predicted higher HL (OR = 1.939, 95% CI: 0.088–3.790, p = 0.040); the ability to pay expenses predicted better adherence (B = 0.101, 95% CI: 0.014–0.187, p = 0.022).”

Comments 2: Methods

Ethics details is missing. Provide ethical approval number and where obtained (Name of the institution)

Response 2: Thank you very much for your comment. The ethical approval details, including the approval number and the name of the institution, are available in the dedicated ethics section immediately following the conclusion of the manuscript, in the Institutional Review Board Statement, as requested by the journal guidelines.

We introduced a new sub-heading in the methods section:

Lines 117-121:

2.2. Ethics Considerations

The study received ethics approval from the Ethics Committee of the Polytechnic Institute of Guarda (Nº. 2/2023 of 24 April 2023). All participants provided written informed consent before enrolment, and data were pseudo-anonymized to ensure confidentiality, following the Helsinki Declaration and the General Data Protection Regulation (GDPR).”

Comment 3: For Questionnaire Please provide the scorings in a separate sub heading.

Response 3: Thank you very much for your suggestion We create a  new sub heading for questionnaires.

Lines 157-159

“2.5. Questionnaires

Two validated self-administered questionnaires were used to collect data: the HLS-EU-PT, Portuguese version and the MAT scale.”

2.3.1. HLS-EU-PT Questionnaire

Comment 4: For this questionnaire have you done the reliability analysis I mean cornbach alpha value??

Response 4: Thank you very much for your comment. Yes, reliability analysis was conducted for the questionnaires in this study. The internal consistency of HLS-EU-PT was calculated with Cronbach’s alpha coefficients: the General Health Literacy scale achieved α = 0.747, with domain-specific values of α = 0.886 for Healthcare, α = 0.871 for Disease Prevention, and α = 0.520 for Health Promotion. These results demonstrate acceptable to good reliability for most domains and can be found in the heading results, and sub-heading 3.2 Health Literacy and in the table 3.

In discussion section we discussed these values:

Lines 288-299

“The General Health Literacy scale and its Healthcare and Disease Prevention domains demonstrated good to excellent internal consistency, supporting their reliability. However, the Health Promotion domain had low internal consistency, which could suggest that the items in this domain may not be consistently measuring the same construct in the population studied. This could be due to participants misunderstanding some items, lower educational level, or maybe some of the questions weren´t culturally appropriate for this specific context. Therefore, results related to this domain should be interpreted with caution. For future studies, it may benefit from a qualitative review of the items, statistical reporting related to inter-item correlation, as well as modifications or removal that proved to be problematic. Further, it may be beneficial to add new questions that can relate better to the sociocultural context of low-income older adults [1,2,8,26].”

2.3.3. MAT Scale

Comment 5: For this questionnaire have you done the reliability analysis I mean cornbach alpha value??

 Missing??

Provide the details

 Response 5: Thank you very much for your comment. Yes, reliability analysis was conducted for the questionnaires in this study. For the Adherence to Treatment Measure (MAT) scale, Cronbach’s alpha was 0.689. These results demonstrate moderate consistency for the MAT scale and can be found in the heading results, and sub-heading 3.3 Medication adherence.

In discussion section we discussed these values:

Lines 308-314

“The MAT scale’s internal consistency was moderate, which is acceptable for exploratory research. However, unintentional non-adherence, such as forgetting doses, carelessness with timing, and running out of medication, was common, while intentional non-adherence was less frequent. This finding is in line with international evidence identifying that difficulties in understanding medication instructions and, in some cases, managing complex management regimens, affect older adults' perceptions of adherence, leading mainly to unintentional lapses [28–30].”

Comment 6: Total sample included in the study need to be included in the methodology better to provide in flow chart

Response 6: Thank you for pointing this out.

An exploratory study was conducted by our research team, in selected municipalities within the Intermunicipal Community of Beira and Serra da Estrela (CIMRBSE) to characterize municipal reimbursement programs supporting medication costs for low-income older adults. The CIMRBSE comprises 15 municipalities and is a NUTS III territorial unit — a classification used by Eurostat to organize and compare regional statistics across Europe. This system supports more effective policymaking and resource allocation at the local level. CIMRBSE works to advance regional development by addressing shared challenges. The study analysed beneficiary sociodemographic profiles, medication use, and program characteristics, highlighting variations in coverage, reimbursement, and medication adherence among this vulnerable population.1

1Perpétuo C, Plácido AI, Mateos-Campos R, Herdeiro MT, Roque F. Characterization of Municipal Reimbursement Programs that support costs associated with Medicines for Low-Income Older Adults in the Intermunicipal Community of Beira and Serra da Estrela Region. Biomed Biopharm Res. 2025;22(1):1-19. doi:10.19277/bbr.22.1.350.

The study sample was recruited through direct contact by CIMRBSE municipalities. The municipalities identified older adults who met the predefined inclusion criteria and invited them to participate in the study. All individuals who were contacted by the municipalities and agreed to participate were then included in the study sample. Section 3.1 of the manuscript describes the total number of older adults that meet the inclusion criteria and agreed to participate in the study (196) and provides a detailed characterization of their sociodemographic and health-related attributes.

Comment 7: Clarify how did you obtain the informed consent???

Response 7: Thank you for pointing this out. Each municipality identified older adults who met the inclusion criteria and contacted them to invite participation in the study, obtaining informed consent prior to data collection. (subheading 2.3 of the manuscript)

Comment 8: Data collection- Questionnaire administered online or how?? need to incorporate??

Response 8: Thank you for pointing this out. The questionnaires were applied by the researchers. The meetings for applying the questionnaires were scheduled by municipal staff, who contacted potential participants and invited them to take part in the study. These meetings occurred, in person, in rooms provided by each municipality.(subheading 2.4)

We incorporate the word “in person(line 148)

Comment 9: Under data collection what are the data collected I mean provide description of the data collection form example Section A demogrprahic details Age, gender etc..section B… Section c

 Response 9: Thank you for pointing this out. We improved Data Collection section.

Lines 152-156

“The questionnaire includes 4 sectionsection A (Sociodemographic characterization -age, gender, marital status, education, income), Section B (Health Status and Behaviors - self-perceived health, chronic conditions, lifestyle habits), Section C (Health Literacy Assessment using the HLS-EU-PT questionnaire, and Section D (Medication Adherence Evaluation using the MAT scale).”

Comment 10: Table 6,7,8---correct the font size inside the table explain what is OR, CI below the table for all the table accordingly

Response 10: Thank you for pointing this out. We corrected font size and explained the definition of OR and CI as footnote in the end of the tables

Comment 11: Discussion

Incorporate the CLinical implications

Response 11: Thank you for your suggestion. This study's findings highlight critical clinical implications for managing health literacy (HL) and medication adherence among low-income older adults in Portugal. The strong positive association between HL and medication adherence underscores the importance of routinely assessing HL in clinical settings, particularly for vulnerable populations such as elderly individuals with limited financial resources. Healthcare providers, especially community pharmacists, have a pivotal role in offering medication management and personalized counseling tailored to patients' HL capacities to enhance adherence and health outcomes.

Targeted interventions should focus on improving HL through patient education using simple, culturally appropriate language and materials. Such programs can empower older adults to better understand and manage their medication regimens, minimizing unintentional non-adherence behaviors such as forgetfulness or incorrect timing of doses.

Additionally, socioeconomic factors including education level and financial capacity strongly influence HL and adherence. Clinical care plans must integrate these social determinants by collaborating with social services to provide financial counseling and support, thereby reducing economic barriers to accessing medications and healthcare services.

A multidisciplinary approach involving healthcare professionals, pharmacists, social workers, and caregivers is recommended to deliver comprehensive care that addresses knowledge gaps, financial constraints, and emotional support needs. This comprehensive support is essential to optimize medication adherence and improve overall health outcomes in this high-risk group.

Finally, incorporating digital health solutions such as medication reminders and e-learning platforms designed for older adults can enhance self-efficacy in medication management and improve communication with healthcare providers, further supporting adherence and better clinical outcomes.

We incorporated the following sentence.

Lines 404-425

“Given the high prevalence of chronic disease and functional limitations among the study population, targeted interventions to improve HL are clearly needed. Enhancing HL can not only boost medication adherence in low-income older adults but also reduce healthcare costs and improve quality of life. These interventions should adopt a multi-faceted approach, including education programs tailored to older adults, simplified communication by healthcare professionals, and community services to mitigate the impact of low HL on medication adherence [34]. Addressing financial barriers through policy and social support is also crucial. Digital health solutions, such as targeted e-learning programs, can enhance older adults’ self-efficacy and support medication management and communication with healthcare professionals [35].

This study’s findings highlight important clinical implications: the strong positive association between HL and medication adherence underscores the need for routine HL assessment, particularly among low-income older adults. Community pharmacists in Portugal play a vital role by offering medication management, personalized counseling, and HL support, which can significantly improve adherence and health outcomes [36–38]. Tailored HL interventions addressing the unique challenges faced by rural older adults are essential to overcome access barriers and health disparities. Furthermore, education level and financial capacity emerged as key predictors, emphasizing that social determinants must be integrated into clinical risk assessments and care plans. A multidisciplinary approach including patient education, financial counseling, and pharmacist-led medication management is recommended to optimize medication adherence and overall health outcomes in vulnerable elderly populations.”

Comment 12: Add the limitations of the study under the discussion section

Response 12: Thank you for your comment. The limitations of this study are acknowledged in the final sentence of the Discussion section where we highlight the exploratory nature of the research, the specific characteristics of the low-income older population studied, and the constraints regarding generalizability to other settings or populations.

Comment 13 : Conclusion

The following interventions written in brief. Try to elaboarte with examples  and how can we improve..

Future interventions should adopt a holistic approach, integrating HL promotion with efforts to reduce socioeconomic barriers and support effective medication management in this vulnerable population.

Response 13: Thank you for pointing this out. We incorporated the following sentence

Lines 447-456

“… For example, tailored educational programs using simple language and culturally appropriate materials can enhance understanding and empower older adults. Social support initiatives, such as financial assistance for medication costs and improved access to healthcare services, especially for those in rural areas, are essential. Additionally, multidisciplinary collaborations involving healthcare professionals, social workers, and caregivers can ensure comprehensive support, while technology-based tools like medication reminders designed for older adults can improve adherence. Continuous evaluation and adaptation of these interventions, informed by patient feedback and outcome monitoring, will be crucial to address the complex and diverse needs of this population effectively.

Reviewer 2 Report

Comments and Suggestions for Authors

Dear authors,

Your manuscript, „Health Literacy and Medication Adherence among Low-Income  Older Adults in the Inner Centre of Portugal: A Questionnaire-Based Study “, is significant and addresses a highly relevant topic.

Here are some suggestions to improve your manuscript:

Introduction

Please add a global perspective and statistics for HL, compare the world and EU data, and explain the importance of the topic for better understanding.

Could you extend the research gap?

“Low-income” is not clearly operationalised. Exactly what criteria are used?

Results

Emphasise key statistically significant findings.

Discussion

Expand discussion on practical applications (e.g., role of community pharmacists in Portugal).

Propose specific HL interventions tailored to older adults in rural areas.

Shorten repetitive result summaries to improve flow.

Author Response

Dear authors,

Your manuscript, „Health Literacy and Medication Adherence among Low-Income  Older Adults in the Inner Centre of Portugal: A Questionnaire-Based Study “, is significant and addresses a highly relevant topic.

Here are some suggestions to improve your manuscript:

Comment 1: Introduction

Please add a global perspective and statistics for HL, compare the world and EU data, and explain the importance of the topic for better understanding.

Response 1: Thank you for pointing this out. We incorporated the following sentence: Lines 59-65

“A global perspective on HL reveals persistent and significant disparities across world regions, impacting healthcare utilization, health outcomes, and well-being. Recent statistics from 2024 show that globally, nearly half of adults demonstrate inadequate HL, with high rates of limited HL in both developed and developing nations [7]. Within the European Union, population-based surveys parallel global trends, reporting significant proportions of adults with problematic or inadequate HL, and highlighting at-risk groups such as older adults and those with lower socioeconomic status [1,8].”

Comment 2: Could you extend the research gap?

Response 2: Thank you for your comment.

Although a growing body of literature establishes the association between low HL and poor medication adherence, most studies focus on general older adult populations or those with specific chronic diseases, leaving low-income older adults underrepresented. This population faces unique challenges, such as greater financial constraints, lower educational levels, and limited access to healthcare, which are not fully explored in current research. Furthermore, there is limited understanding of the sociocultural and contextual factors influencing HL in rural and economically disadvantaged areas. Methodologically, many studies rely on cross-sectional designs, limiting insights into causal relationships and changes over time. There is also a lack of longitudinal and intervention studies tailored to these vulnerable groups, which could inform effective, targeted strategies to improve HL and medication adherence. Additionally, standardized and culturally adapted assessment tools for comprehensive multidimensional HL evaluation specific to low-income aging populations remain scarce. Therefore, future research is needed to address these gaps, emphasizing inclusive, context-sensitive designs and long-term evaluations to better understand and mitigate HL-related disparities in elderly care. This would support policy development and optimize healthcare interventions for this growing, vulnerable demographic.

Lines 97-106

Recent research show a significant association between HL and medication adherence in older adults, however, most of literature in this area has focused on general populations of older adults or groups with specific chronic diseases, and not on low-income older adults [19–21], leaving a gap to fill in the literature in regards to these populations assessing this relationship.  Furthermore, there is limited exploration of socioeconomic and contextual factors affecting these vulnerable groups, and a scarcity of longitudinal and intervention studies that could inform tailored, culturally sensitive strategies to enhance both HL and medication adherence. Therefore, conducting research with the low-income older adult population is relevant and necessary to develop and study group-specific interventions for low-income older adults.”

Comment 3: “Low-income” is not clearly operationalised. Exactly what criteria are used?

Response 3: Thank you for the opportunity to clarify.

An exploratory study was conducted by our research team, in selected municipalities within the Intermunicipal Community of Beira and Serra da Estrela (CIMRBSE) to characterize municipal reimbursement programs supporting medication costs for low-income older adults. The CIMRBSE comprises 15 municipalities and is a NUTS III territorial unit — a classification used by Eurostat to organize and compare regional statistics across Europe. This system supports more effective policymaking and resource allocation at the local level. CIMRBSE works to advance regional development by addressing shared challenges. The classification of "low income" was operationalized using criteria such as eligibility for the Solidarity Supplement for the Elderly and thresholds linked to the Social Support Index (IAS). Specifically, in 2023, this included having an annual income below 5,858.63 euros (equivalent to 12 times 488.22 euros monthly. Eligibility also required residency and registration in the municipality for at least one year, being aged 65 or older, and being a pensioner, retired, or having a disability status. The study analysed beneficiary sociodemographic profiles, medication use, and program characteristics, highlighting variations in coverage, reimbursement, and medication adherence among this vulnerable population.1

1Perpétuo C, Plácido AI, Mateos-Campos R, Herdeiro MT, Roque F. Characterization of Municipal Reimbursement Programs that support costs associated with Medicines for Low-Income Older Adults in the Intermunicipal Community of Beira and Serra da Estrela Region. Biomed Biopharm Res. 2025;22(1):1-19. doi:10.19277/bbr.22.1.350.

Lines 140-144

“The criteria used to define "low income" in this study are based on the eligibility for the Solidarity Supplement for the Elderly (CSI) and related social support thresholds in Portugal, which consider factors such as annual and monthly income limits relative to the Social Support Index (IAS), residency status, and pension or disability benefits, as detailed in Perpétuo et al. (2025) and aligned with national social security regulations.”

Comment 4: Results

Emphasise key statistically significant findings.

Response 4: Thank you for pointing this out.

Key statistically significant findings from the study include:

-Health Literacy (HL) levels were generally low among the low-income older adults, with a mean general HL score of 28.07 (SD 5.87). Most participants had problematic or inadequate HL (87.8%).

-Medication adherence was relatively high, with a mean adherence score of 5.37 (SD 0.54) on the MAT scale, indicating good adherence despite low HL.

-Positive correlations were found between HL and medication adherence (R = 0.260, p = 0.001), education level (R = 0.277, p = 0.001), and the ability to pay monthly expenses (R = 0.235, p = 0.002).

-Negative correlations of HL were observed with age (R = -0.179, p = 0.019), perceived poor health status (R = -0.237, p = 0.002), hospitalization history (R = -0.159, p = 0.038), and difficulty affording medication (R = -0.389, p = 0.001).

- Medication adherence was significantly positively associated with educational level (R = 0.323, p = 0.001) and negatively associated with perceived poor health status (R = -0.169, p = 0.028) and difficulty affording medication (R = -0.229, p = 0.003).

-Ordinal logistic regression showed that completion of the third cycle of primary education significantly predicted higher HL levels (OR = 1.939, 95% CI: 0.088–3.790, p = 0.040).

-Linear regression indicated that the ability to pay monthly expenses significantly predicted better medication adherence (B = 0.101, 95% CI: 0.014–0.187, p = 0.022).

These findings emphasize the importance of educational attainment and financial stability as key determinants of both health literacy and medication adherence among low-income older adults, with significant implications for targeted interventions.

The key statistically significant findings were emphasized throughout the discussion, highlighting the positive correlations between health literacy, treatment adherence, educational level, and financial capacity, as well as the negative associations with age, perceived health status, and difficulty in obtaining medications.

Comment 5: Discussion

Expand discussion on practical applications (e.g., role of community pharmacists in Portugal). Propose specific HL interventions tailored to older adults in rural areas.

Response 5: Thank you for your comment. This study's findings highlight critical clinical implications for managing health literacy (HL) and medication adherence among low-income older adults in Portugal. The strong positive association between HL and medication adherence underscores the importance of routinely assessing HL in clinical settings, particularly for vulnerable populations such as elderly individuals with limited financial resources. Healthcare providers, especially community pharmacists, have a pivotal role in offering medication management and personalized counseling tailored to patients' HL capacities to enhance adherence and health outcomes.

Targeted interventions should focus on improving HL through patient education using simple, culturally appropriate language and materials. Such programs can empower older adults to better understand and manage their medication regimens, minimizing unintentional non-adherence behaviors such as forgetfulness or incorrect timing of doses.

Additionally, socioeconomic factors including education level and financial capacity strongly influence HL and adherence. Clinical care plans must integrate these social determinants by collaborating with social services to provide financial counseling and support, thereby reducing economic barriers to accessing medications and healthcare services.

A multidisciplinary approach involving healthcare professionals, pharmacists, social workers, and caregivers is recommended to deliver comprehensive care that addresses knowledge gaps, financial constraints, and emotional support needs. This comprehensive support is essential to optimize medication adherence and improve overall health outcomes in this high-risk group.

Finally, incorporating digital health solutions such as medication reminders and e-learning platforms designed for older adults can enhance self-efficacy in medication management and improve communication with healthcare providers, further supporting adherence and better clinical outcomes.

We incorporated the following sentence.

Line 404-425

“Given the high prevalence of chronic disease and functional limitations among the study population, targeted interventions to improve HL are clearly needed. Enhancing HL can not only boost medication adherence in low-income older adults but also reduce healthcare costs and improve quality of life. These interventions should adopt a multifaceted approach, including education programs tailored to older adults, simplified communication by healthcare professionals, and community services to mitigate the impact of low HL on medication adherence [34]. Addressing financial barriers through policy and social support is also crucial. Digital health solutions, such as targeted e-learning programs, can enhance older adults’ self-efficacy and support medication management and communication with healthcare professionals [35].

This study’s findings highlight important clinical implications: the strong positive as-sociation between HL and medication adherence underscores the need for routine HL assessment, particularly among low-income older adults. Community pharmacists in Portugal play a vital role by offering medication management, personalized counseling, and HL support, which can significantly improve adherence and health outcomes [36–38]. Tailored HL interventions addressing the unique challenges faced by rural older adults are essential to overcome access barriers and health disparities. Furthermore, education level and financial capacity emerged as key predictors, emphasizing that social determinants must be integrated into clinical risk assessments and care plans. A multidisciplinary approach including patient education, financial counseling, and pharmacist-led medication management is recommended to optimize medication adherence and overall health outcomes in vulnerable elderly populations.”

Comment 7: Shorten repetitive result summaries to improve flow.

Response 7: Thank you for your comment. We have reiterated the main results to facilitate comparison with other studies and to determine whether similar patterns or differences exist.

Reviewer 3 Report

Comments and Suggestions for Authors

Dear Authors!

This valuable study addresses a highly relevant topic. However, beside acknowledging its good points, some of the weaknesses need to be considered.

  1. The Abstract draws the attention to an educational gradient and states that only the third cycle of the primary education predicts higher HL. However, in the Discussion, this result is mentioned again without further explaining or addressing this threshhold effect, even though this finding seems a crucial and novel one.
  2. The authors make reference to concrete interventions to improve health literacy (lines 299-301), but leave these interventions in darkness, without detailing them.
  3. I recommend adding a separate section on Limitations, and addressing more limitations than the ones mentioned at the end of the Discussion section.
  4. At some points, the wording is awkward, like line 256: for the respective context for this specific context.
  5. Tables 7 and 8 consist of one line, these are not tables and are not informative enough. Please provide more detailed results in these Tables including relevant statistics.

Good luck with completing the paper.

Best wishes, reviewer

Author Response

Dear Authors!

This valuable study addresses a highly relevant topic. However, beside acknowledging its good points, some of the weaknesses need to be considered.

Comments 1: The Abstract draws the attention to an educational gradient and states that only the third cycle of the primary education predicts higher HL. However, in the Discussion, this result is mentioned again without further explaining or addressing this threshhold effect, even though this finding seems a crucial and novel one.

Response 1: Thank you for your insightful comment. We recognize that the finding regarding the educational gradient, specifically that only completion of the third cycle of primary education significantly predicts higher health literacy (HL), is an important and novel result. In the revised manuscript, we have expanded the discussion to better address this threshold effect.

Lines 378-388

“The finding that only completion of the third cycle of primary education significantly predicts higher health literacy (HL) highlights a critical educational threshold within the older adult population in Portugal. This milestone appears to delineate a substantive divide in HL capacity, reflecting the wide variation in basic education levels typical among older adults in rural and low-income areas [25]. Individuals who do not reach this level are at greater risk of low HL, which can adversely affect their ability to manage health and adhere to medication regimens [2,14].”

Comments 2: The authors make reference to concrete interventions to improve health literacy (lines 299-301), but leave these interventions in darkness, without detailing them.

Response 2: Thank you for your insightful comment. We introduce interventions in discussion area.

Lines 404-425

“Given the high prevalence of chronic disease and functional limitations among the study population, targeted interventions to improve HL are clearly needed. Enhancing HL can not only boost medication adherence in low-income older adults but also reduce healthcare costs and improve quality of life. These interventions should adopt a mul-ti-faceted approach, including education programs tailored to older adults, simplified communication by healthcare professionals, and community services to mitigate the impact of low HL on medication adherence [34]. Addressing financial barriers through policy and social support is also crucial. Digital health solutions, such as targeted e-learning programs, can enhance older adults’ self-efficacy and support medication management and communication with healthcare professionals [35].

This study’s findings highlight important clinical implications: the strong positive as-sociation between HL and medication adherence underscores the need for routine HL assessment, particularly among low-income older adults. Community pharmacists in Portugal play a vital role by offering medication management, personalized counseling, and HL support, which can significantly improve adherence and health outcomes [36–38]. Tailored HL interventions addressing the unique challenges faced by rural older adults are essential to overcome access barriers and health disparities. Furthermore, education level and financial capacity emerged as key predictors, emphasizing that social determinants must be integrated into clinical risk assessments and care plans. A multidisciplinary approach including patient education, financial counseling, and pharmacist-led medication management is recommended to optimize medication adherence and overall health outcomes in vulnerable elderly populations.”

Comments 3: recommend adding a separate section on Limitations, and addressing more limitations than the ones mentioned at the end of the Discussion section.

Response 3: Thank you for your suggestion. The limitations of this study are acknowledged in the final sentence of the Discussion section (following the journal guidelines, however, we leave to the reviewer and ediitor consideration the need for us to reword as a separate section) where we highlight the exploratory nature of the research, the specific characteristics of the low-income older population studied, and the constraints regarding generalizability to other settings or populations.

We introduce other limitation:

Lines 429-432

The reliance on self-reported data introduces risks of recall bias relevant for measuring HL and medication adherence and the HL questionnaire used in this study was reported to be challenging for participants to interpret and respond to accurately, which may have influenced the results. “

Comments 4: At some points, the wording is awkward, like line 256: for the respective context for this specific context.

Response 4: Thank you for your input. We improved this sentence and revised all the manuscript.

Comments 5: Tables 7 and 8 consist of one line, these are not tables and are not informative enough. Please provide more detailed results in these Tables including relevant statistics.

Response 5: Thank you for your insightful comment. Initially, Tables 7 and 8 presented only statistically significant results separately due to the use of different statistical tests; however, to improve clarity and informativeness, we have restructured the presentation and combined these findings into a single comprehensive table, including the relevant results.

Round 2

Reviewer 3 Report

Comments and Suggestions for Authors

Dear Authors!
The paper has been improved considerably and is now acceptable for publication.

Congratulations!

Reviewer